# Reversible Adhesive Bio-Toe with Hierarchical Structure Inspired by Gecko

**DOI:** 10.3390/biomimetics8010040

**Published:** 2023-01-16

**Authors:** Liuwei Wang, Zhouyi Wang, Bingcheng Wang, Qingsong Yuan, Zhiyuan Weng, Zhendong Dai

**Affiliations:** 1College of Mechanical and Electrical Engineering, Nanjing University of Aeronautics and Astronautics, Nanjing 210016, China; 2Shenzhen Research Institute, Nanjing University of Aeronautics and Astronautics, Shenzhen 518063, China

**Keywords:** bio-toe, reversible adhesion, hierarchical structure, gecko

## Abstract

The agile locomotion of adhesive animals is mainly attributed to their sophisticated hierarchical feet and reversible adhesion motility. Their structure–function relationship is an urgent issue to be solved to understand biologic adhesive systems and the design of bionic applications. In this study, the reversible adhesion/release behavior and structural properties of gecko toes were investigated, and a hierarchical adhesive bionic toe (bio-toe) consisting of an upper elastic actuator as the supporting/driving layer and lower bionic lamellae (bio-lamellae) as the adhesive layer was designed, which can adhere to and release from targets reversibly when driven by bi-directional pressure. A mathematical model of the nonlinear deformation and a finite element model of the adhesive contact of the bio-toe were developed. Meanwhile, combined with experimental tests, the effects of the structure and actuation on the adhesive behavior and mechanical properties of the bio-toe were investigated. The research found that (1) the bending curvature of the bio-toe, which is approximately linear with pressure, enables the bio-toe to adapt to a wide range of objects controllably; (2) the tabular bio-lamella could achieve a contact rate of 60% with a low squeeze contact of less than 0.5 N despite a ±10° tilt in contact posture; (3) the upward bending of the bio-toe under negative pressure provided sufficient rebounding force for a 100% success rate of release; (4) the ratio of shear adhesion force to preload of the bio-toe with tabular bio-lamellae reaches approximately 12, which is higher than that of most existing adhesion units and frictional gripping units. The bio-toe shows good adaptability, load capacity, and reversibility of adhesion when applied as the basic adhesive unit in a robot gripper and wall-climbing robot. Finally, the proposed reversible adhesive bio-toe with a hierarchical structure has great potential for application in space, defense, industry, and daily life.

## 1. Introduction

Mimicry of organisms in nature has inspired many significant inventions that have influenced the course of human civilization [1]. Animals have evolved pedipalps that form adhesive contact with environmental surfaces and give them the ability to regulate adhesion when foraging and escaping from natural predators, and the gecko is a good representative [2]. By investigating the adhesion mechanisms of geckos, researchers expect to develop bionic adhesion mechanisms/robots to operate under complex and challenging working conditions [3]. With in-depth understanding of the gecko’s hierarchical adhesion system and rapid progress in micro-nano fabrication/growth technologies, dry-adhesive surfaces with microstructures similar to gecko bristles have been developed [4]. By optimizing the structure and material of dry-adhesive surfaces, their adhesion performance can meet or surpass that of the gecko itself. Grippers using adhesion technology are capable of grasping smooth surfaces, fragile components, and soft objects that are difficult to grasp by conventional frictional grippers, which can be widely used in industrial assembly and intelligent agriculture [5,6]. In the space environment, adhesion mechanisms can reliably attach targets with a wide range of linear and rotational velocities, laying the foundation for spacecraft maintenance, noncooperative target capture, and other operations [7]. Wall-climbing robots with dry-adhesive feet can climb various slopes [8,9], providing a good mobile platform for counterterrorism reconnaissance and disaster relief. However, the performance of adhesion mechanisms/robots still falls short of expectations, and the advantages of bionic dry adhesion have not been fully exploited in engineering applications.

The toe of the gecko has a precise hierarchical structure from the microscopic to macroscopic scale (spatula–seta–lamella–toe). It is widely accepted that van der Waals forces generated by the close contact between microscopic spatula pads and environmental surfaces are the main sources of adhesion [10]. Mechanical tests on the single seta, setae arrays, and individual toe suggest that the reliable and rapid switching of the gecko’s adhesion and release largely depends on the control of the attachment/detachment angle [11]. In the natural configuration, the setae shafts are at an angle of approximately 30° to the substrate and do not generate adhesion forces. During adhesion, the toes roll in and shearly pull the setae to reduce the contact (attachment) angle, resulting in an adhesion force of 40 μN and a friction force of 200 μN for a single seta [12]. During release, the toes roll out, and the setae relax back to their initial free (natural) state with a significant decrease of more than three orders of magnitude in the adhesion and friction forces [13]. Moreover, the slender geometry and asymmetric structure contribute to quick adhesion and detachment [14]. The gecko-inspired geometry with a tilted pad attached to a curved beam can realize on/off switching of adhesion, with lateral sliding in addition to vertical debonding [15]. Soft lamellae act as a soft spring to sustain most of the normal deformation during preloading and maintain a wide range of adhesive states rather than a repulsive state [16,17]. In general, the hierarchical structural properties and flexible reversible adhesion/release behavior (rolling in/out) significantly contribute to the agile locomotion ability of geckos [18]. Therefore, for engineering applications of biomimetic adhesion technology, it is essential to fully integrate the lower adhesive layer and the upper supporting/driving layer of the adhesion mechanism from a systemic perspective.

The adhesive devices for transferring flat targets consist of several flat adhesion units with wedge-shaped microstructured dry-adhesive surfaces. The adhesion units are symmetrically placed and realize the adhesion/release of delicate and fragile flat objects, such as silicon wafers, glass wafers, and circuit boards, by shear loading/unloading. However, the rigid supporting layer limits the adhesion range of the unit to flat surfaces, and the shear-driving mechanism complicates the adhesive device [6,19,20,21]. Similarly, flat adhesion units have also been used in the feet of wall-climbing robots. These units widen the range of climbing angles, but do not allow the robot to climb curved surfaces [9,22,23]. Adhesion devices using tendon-driven underactuated structures [5,24,25,26], fluid-driven elastic actuators [27,28], or fabric bending actuators [29] as supporting and driving layers exhibit good drive flexibility and interaction safety. They can envelope and adhere to an extensive range of targets. However, owing to the nonlinear mechanical properties of the material and fluid drive, the mechanical response of the aforementioned soft adhesion devices is difficult to characterize accurately, reducing the controllability and stability of the adhesion. In addition, the excellent ability to adapt to objects does not guarantee the interfacial fit of the contact area [30], and the structural design for optimizing the interfacial adhesive contact has rarely been investigated [31]. In general, research on the biomimetic adhesive unit requires investigation of the geckos’ hierarchical adhesion system, clarification of the relationships between the upper supporting/driving layer and the lower adhesive layer (effects of the structural design, material characteristics, and driving property of the supporting/driving layer on the adhesion performance), and structure–drive–function integration of the adhesive unit.

In this work, the gecko toe’s reversible adhesive behavior of rolling in/out and multilayered structure properties are discussed. The design of a reversible bionic toe (bio-toe) with a hierarchical structure is proposed based on biological inspiration. The adhesive behavior and mechanical properties of the bio-toe are analyzed by establishing mathematical and finite-element models. Combined with experimental tests, the effects of the structural properties and actuation on the adhesive performance of the bio-toe are investigated. Adhesive mechanisms based on the bio-toe for robotic grippers and wall-climbing robots are developed, and functional tests were performed to promote the application of bionic dry-adhesive technology.

## 2. Design and Analysis

### 2.1. Upper Supporting/Driving Layer: Elastic Actuator

#### 2.1.1. Design of the Structure and Mathematical Model

The adhesion/release behavior of the gecko toe, captured by a high-speed camera (Figure 1a), shows that the toe bends downward until it adhered to the surface stably during adhesion and bends upward until it entirely separates from the surface during release. In general, a flexible toe with high freedom and bidirectional bending ability is the biological basis for the reversible adhesion of the gecko, and the muscle stretch provides the driving force for reliable adhesion and rapid release, respectively. Therefore, for adhesive bio-toes, flexible and controlled downward/upward bending is required for reversible adhesion and high adaptability, as well as sufficient structural stiffness and driving force to maintain stable adhesion and effective release.

In engineering, mechanisms capable of controllable bending deformation mainly include electroactive polymers (IPMCs, DEAs, etc.), motor-driven joint mechanisms, and fluid-driven elastic actuators [32]. Among these, electroactive polymers usually have high requirements for the working environment and drive mode and are deficient in deformation rate and load capacity. Joint mechanisms have high strength and working accuracy, but they have low fault tolerance and are complex in structure and control. Fluid-driven elastic actuators have advantages in terms of flexibility, adaptability, interaction safety, and system simplicity, but are inferior to joint mechanisms in terms of operating accuracy and output forces. Because the reversible adhesion/release process is fast, the bio-toe should be flexible and have a high fault tolerance. In addition, because adhesion contact requires much less interfacial squeezing than friction [26], high output forces are not required. Therefore, the fluidic-driven elastic actuator shown in Figure 1b was designed as the toe’s upper supporting/driving layer, which consists of four internally connected square cavities and an abdomen. Owing to the limitation of the abdomen, the elastic actuator can bend upward and downward under bidirectional pressure. The initial state of the bio-toe is upward. This design ensures that when the bio-toe adheres to the flat substrate, there is a positive pressure inside to provide a preload for steady adhesion. In addition, it also increases the upward bending deformation of the bio-toe under negative pressure, which is conducive to the dry-adhesive surface reaching the critical detachment angle, contributing to rapid release from the substrate.

Inspired by the mathematical modeling method proposed by Hao et al. [33], a mathematical model that captures both the geometrical properties of the elastic actuator and the applied pressure was established to describe the deformation (Figure 1d) of the bio-toe. To simplify the model and ignore some minor factors, we made a few simplifications: (1) the length of the abdomen was assumed to be constant; (2) the bio-toe deforms mainly by expanding or contracting the cavities, and the tensile deformation of the cavity surface is ignored; (3) the bending curvature of a single cavity is assumed to be the same as that of the bio-toe. Because the four cavities have the same geometry, we only analyzed the bending curvature of a single cavity (Figure 1d). For a single cavity, the axial (*x*-axis) stress $\sigma_{x}$ and radial (*y*-axis) stress $\sigma_{y}$ are as follows:(1)$$\sigma_{x}=\frac{P(wh_{1}-\pi r^{2}/2)\cos(\theta_{1}/2)}{2d_{2}h_{1}+d_{2}(w+2d_{2})}$$
(2)$$\sigma_{y}=\frac{Pw(R-d_{1}-h_{2})}{2d_{2}(R-d_{1})}$$
where P is the air pressure applied to the inner wall of the cavity, $h_{1}$ is the height of the cavity, $h_{2}$ is the height of the airway, w is the width of the cavity, $d_{1}$ is the thickness of the inextensible abdomen, $d_{2}$ is the thickness of the cavity, R is the radius of the elastic actuator, r is the radius of the rib, $\theta_{1}$ is the fillet angle corresponding to a single cavity, and $\theta_{2}$ is the fillet angle corresponding to a single rib (Figure 1c).

The deformation of a single cavity along the axial and radial directions is
(3)$$d_{x}=l_{\overset{⌢}{P_{1}P_{2}}\_x}\varepsilon_{x}(\sigma_{x})$$
(4)$$d_{y}=l_{P_{1}P_{3}\_y}\varepsilon_{y}(\sigma_{y})$$
where $l_{\overset{⌢}{P_{1}P_{2}}\_x}$ is the length of the projection of $\overset{⌢}{P_{1}P_{2}}$ on the *x*-axis, $l_{P_{1}P_{3}\_y}$ is the length of the projection of *P*_1_*P*_3_ on the *y*-axis, $\varepsilon_{x}$ is a function of $\sigma_{x}$, and $\varepsilon_{y}$ is a function of $\sigma_{y}$, where
(5)$$l_{\overset{⌢}{P_{1}P_{2}}\_x}=2(R-d_{1}-h_{2}-d_{2})\sin[(\theta_{1}+\theta_{2})/2]$$
(6)$$l_{P_{1}P_{3}\_y}=(h_{1}/2+h_{2}+r+d_{2})\cos[(\theta_{1}+\theta_{2})/2]$$

The relationship between ε and σ can be obtained by calibrating the finite element analysis (FEA) results. Here, we selected a set of specific elastic actuator parameters, obtained the function ε(σ) to calculate the bending curvature of the actuator, and compared the results with those of physical experiments to verify the model. To get the function, we obtained the strain of the single cavity on the *x* and *y* axes under different pressures in the simulation and then fit the data with quadratic polynomial functions as follows:(7)$$\varepsilon_{x}=3E-05\sigma_{x}{}^{2}+0.0018\sigma_{x}+0.0115\begin{array}{cc} & \left(R^{2}=0.9993\right) \end{array}$$
(8)$$\varepsilon_{y}=3E-05\sigma_{y}{}^{2}+0.0014\sigma_{y}-0.0012\begin{array}{cc} & \left(R^{2}=0.9999\right) \end{array}$$

Figure 1e shows the fitness of the data functions. We defined the bending curvature of the single cavity as less than, equal to, and greater than zero in the upward, horizontal, and downward bending states, respectively.

By assumption, the length of $\overset{⌢}{P_{3}P_{4}}$ remains constant. According to the Figure, the center angles of $\overset{⌢}{P_{1}P_{2}}$ and $\overset{⌢}{P_{3}P_{4}}$ after expansion are the same, and defining C as the curvature after the deformation of m, the bending curvature *C* is given by
(9)$$\frac{l_{\overset{⌢}{P_{1}{}^{’}P_{2}{}^{’}}}}{C^{-1}+(d_{1}/2+h_{2}+r+d_{2})}=\frac{l_{\overset{⌢}{P_{3}{}^{\prime }P_{4}{}^{\prime }}}}{C^{-1}}$$
where
(10)$$l_{\overset{⌢}{P_{1}{}^{’}P_{2}{}^{’}}}=(R-d_{1}/2-h_{2}-r-d_{2})(\theta_{1}+\theta_{2})+d_{x}$$
(11)$$l_{\overset{⌢}{P_{3}{}^{\prime }P_{4}{}^{\prime }}}=l_{\overset{⌢}{P_{3}P_{4}}}=(R-d_{1}/2)(\theta_{1}+\theta_{2})$$

The bending curvature *C* of the single cavity/bio-toe is
(12)$$C=\frac{l_{\overset{⌢}{P_{1}{}^{’}P_{2}{}^{’}}}-l_{\overset{⌢}{P_{3}P_{4}}}}{l_{\overset{⌢}{P_{3}P_{4}}}(d_{1}/2+h_{2}+r+d_{2})}$$

To verify the accuracy of the mathematical model, Figure 1f compares the bending curvatures of the elastic actuator with the structural parameters in Table 1 versus the pressure for the mathematical model, simulation, and experimental results (the simulation and experimental results are presented in Section 2.1.2 and Section 4.1, respectively). All three results agree well, indicating that the bending curvature of the elastic actuator increases approximately linearly with increasing air pressure. In general, this mathematical model can qualitatively analyze the effects of structural parameters on the bending curvature of an elastic actuator. According to the model, the wider and higher the cavity and the thinner the wall, the better the bending performance of the elastic actuator.

#### 2.1.2. FEA Simulation of Elastic Actuators

The mathematical model only describes the deformation of the elastic actuator but not the output force. Therefore, we systematically investigated the effects of the structural parameters and applied pressures on the bidirectional bending deformation and contact force of the elastic actuator through FEA. First, a set of specific elastic actuator parameters was selected as the basis (Table 1). Second, the effects of *h*_1_, *w*, *d*_1_, and *d*_2_ on the bending curvature *C* and contact force of the elastic actuator under different pressures were simulated and analyzed using the control variable method. Figure 2a shows the finite element models (FEMs) of bending deformations and contact forces of elastic actuators under pressure driving. The Mooney–Rivlin model that can accurately describe the mechanical properties of nonlinear flexible structures similar with the elastic actuator was used [30,34].

The simulation results (Figure 2b) show that the bending curvatures and contact forces of elastic actuators increase with increasing air pressure, with the structural parameters affecting only the increment. The wider the cavity, the higher the cavity, and the thinner the wall thickness, the larger the increase in bending curvature *C* and contact force. According to Section 2.1.1, the bio-toe should be sufficiently flexible to conform to different curvature surfaces and sufficiently strong to preload for stable interfacial adhesion. Therefore, elastic actuators with low cavity widths, heights, and high wall thicknesses were excluded. However, a large cavity width and height directly increase the size and fabrication difficulty of the elastic actuator [34] and the sensitivity of the bending deformation and contact force to the change in pressure, which increases the control difficulty. In general, the structural parameters of the elastic actuator were determined to be identical to those listed in Table 1.

### 2.2. Lower Adhesive Layer: Bio-Lamella

#### 2.2.1. Design of Three Types of Bio-Lamellae

Dry adhesion has high requirements for the contact state (high contact rate and low contact force). As shown in Figure 3a, the lamellae of the gecko toe are discrete and uniformly distributed, which helps to avoid the rapid expansion of local adhesion failure to the entire adhesion area and enhances the adhesion stability. The longitudinal frozen slicing of the lamellae shows that they are in a tabular shape with an angle of 20° to 35° to the abdomen of the toe, giving the lamellae a soft spring-like property that enhances contact adaptation and reduces contact impacts [17]. The bottom view of the lamellae suggests that the end profile of the lamellae can be approximated as a circular arc with a radius of approximately 1.7–2 times the radial width of the toe, and it was proposed that the gecko pulls the curved lamellae with tendons to control the contact angle for reversible adhesion [13]. Therefore, we designed three types of bionic lamellae (bio-lamellae): square, tabular, and curved bio-lamellae (Figure 3b). The bio-toe has three bio-lamellae that act as an adhesive layer. Mushroom-shaped microstructured adhesive surfaces [35] are attached to the outer side of the bio-lamellae (Figure 4a).

#### 2.2.2. FEA Simulation of Bio-Lamellae

The behavior and mechanical characteristics of the three bio-lamellae in contact with a flat substrate were also explored in the simulation. Taking the tabular bio-lamella as an example, three types of FEMs of the tabular bio-lamella under three contact modes (vertically, 10° tilted along the lamella, and 10° tilted against the lamella) were established (Figure 3c). The simulation result for the contact force and contact area versus the load distance of the square bio-lamella is shown in Figure 3d. Because of the axially symmetrical structure of the square bio-lamella, the simulation results of contact at 10° tilted along and against the lamella are the same. When contacting vertically, the contact area rises rapidly to a peak of 260 mm^2^ (100% contact rate) within 0.05 mm of load distance and then remains stable. The contact force also increases sharply to approximately 30 N and continues to increase with increasing load distance. However, when the square bio-lamella makes contact of 10° tilted along (against) the lamella, the area decreases sharply relative to the vertical contact. When the contact force is close to 100 N at a load distance of 1 mm, the contact area is 160 mm^2^ (60% contact rate). The simulation results for the contact characteristics of the tabular and curved bio-lamellae are shown in Figure 3e and Figure 3f, respectively, with similar trends. The contact force of the two bio-lamellae under the three contact modes is stable at less than 0.5 N, increases slowly (we refer to this interval as “low squeeze contact”), and then increases rapidly to more than 10 N with increasing load distance. The distances of low squeeze contact are approximately 4.5, 3.5, and 2.5 mm for the two bio-lamellae making contact against the lamella, vertically, and along the lamella, respectively. The contact area initially remains almost constant and increases rapidly and approximately linearly toward the end of the low-squeeze contact. At the end of the low-squeeze contact, the contact areas of the tabular and curved bio-lamellae reach 150 mm^2^ (58% contact rate) and 50 mm^2^ (19% contact rate), respectively. Subsequently, as the contact force increases rapidly, the contact area increases at a similar rate, until it plateaus. The final steady contact areas for the tabular and curved bio-lamellae are 250 mm^2^ (96% contact rate) and 85 mm^2^ (30% contact rate), respectively.

The simulation results indicate that the square bio-lamella can achieve complete contact more quickly than the other two bio-lamellae when in vertical contact. However, a slight tilt in the contact posture can easily cause a sudden drop in the contact area and a small distance (less than 0.1 mm) of the low squeeze contact, indicating low contact robustness. The tabular and curved bio-lamellae achieve an average low squeeze contact of approximately 3 mm with tilted contact postures. The tabular bio-lamella achieves a contact rate of nearly 60% at low preloads below 0.5 N, implying high contact adaptability and robustness. However, the curved bio-lamella, which is morphologically closest to the biological lamella, has only one-third of the adhesive contact rate of the tabular bio-lamella. The reason for this can be found in the simulation (Figure 3f), as the curved bio-lamella is not adequately contacted because of reverse warping in the middle part when it is in contact with the flat substrate. In general, the tabular bio-lamella has a longer distance from the low-squeeze contact than the other two bio-lamellae, allowing a larger part of the lamella to fit the substrate with a lower contact force. In addition, the tabular bio-lamella is more robust to changes in posture and disturbances in the external environment.

### 2.3. Hierarchical Adhesive Bio-Toes

The bio-toe consists of an elastic actuator and three bio-lamellae. The elastic actuator acts as the supporting/driving layer, and the bio-lamellae cemented with a dry-adhesive surface [35] acts as the adhesive layer. To investigate the effects of the structural differences of the bio-lamellae on the adhesive contact performance of the bio-toe, four types of bio-toes were designed (Figure 4a): an elastic actuator without bio-lamellae (bio-toe1), with three square bio-lamellae (bio-toe2), with three tabular bio-lamellae (bio-toe3), and with three curved bio-lamellae (bio-toe4). FEMs of the four bio-toes adhering to and shearly pulling off from the flat surface were developed to analyze their contact behavior and mechanical properties, and the bio-toe3 is taken as an example in Figure 4b. Figure 4b,c show the two simulation steps (adhesion and shear pull-off) and contact properties of bio-toe3, respectively. During adhesion, bio-toe3 bends downward and adheres to a fixed substrate under positive pressure. At this point, the bio-toe applies a shear forward force and normal preload to the substrate, and the contact area increases rapidly from zero to a steady value. During shear pull-off, the bio-toe contacts tightly with the substrate, and the shear adhesion force increases rapidly and approximately linearly, whereas the normal contact force decreases very slowly. When the adhesion limit is reached, both the contact forces and area rapidly decrease. Figure 4d shows the contact area of the bio-toe in the adhesion step versus the applied pressure. The contact areas of all four toes shows an increasing and then decreasing trend with increasing pressure. The area of bio-toe3 is the highest throughout, increasing from 17 mm^2^ at 20 kPa to a peak of 128 mm^2^ at 50 kPa and dropping back to 78.9 mm^2^ at 0.07 MPa. The contact area of bio-toe2 is the second highest throughout, reaching a peak of 57.4 mm^2^ at 50 kPa. The contact areas of bio-toe1 and 2 reach peaks of 41.2 and 25 mm^2^ at 40 kPa, respectively. The greatest difference is at 50 kPa, where the contact areas of bio-toe3 are 91.2, 106.6, and 39.7 mm^2^ more than those of bio-toe1, 2, and 4, respectively. The maximum shear adhesion force of the bio-toe when pulling off from the substrate is positively correlated with the contact area. In general, the simulation results further validate that the tabular and curved bio-lamellae have better contact adaptability and can improve the contact rate of the bio-toes.

## 3. Experimental Setup

A synchronous testing platform for the adhesive contact state and mechanics was built to evaluate the deformation and adhesion properties of the bio-toe. A schematic of the platform is shown in Figure 5a. Two two-axis force sensors (force range: ±100 N; resolution: 0.1% Fs) were fixed parallel to each other at a distance of 140.0 ± 5.0 mm on an aluminum plate. Two ends on the long side of a transparent acrylic panel (150 × 100 × 4 mm) that acts as the adhered substrate are fixed to the top of each sensor to form a gantry-like two-axis force testing platform. An LED strip was then wound around the acrylic panel. According to the principle of frustrated total reflection [36], the area at which the bio-toe contacts the substrate generates a facula with significantly higher brightness than the non-contact area (Figure 5c), which is convenient for extracting the contact area by computer graphics processing. A flat # 1 mirror was mounted under the acrylic panel at an angle of 45°, and another # 2 mirror was mounted on the side of the acrylic panel at an angle of 45° to its transverse symmetry. A high-speed camera (FLIR, Portland, Oregon, USA) was placed perpendicular to the longitudinal symmetry of the acrylic panels. As shown in Figure 5c, the high-speed camera recorded the distribution of the contact area from mirror #1 and the deformed state of the bio-toe # 2 mirror. The bio-toe was fixed to a two-axis mobile platform and could move freely on the laterally symmetrical surface of the acrylic panel. Markers were drawn with a black pen on the abdominal side of the bio-toe, and the coordinate information of the markers was extracted by computer graphics processing to calculate the bending curvature. The prototypes of the four bio-toes (Figure 5d) were fabricated by the “simultaneous molding and assembly” method [34] using a PDMS silicone rubber material. The elastic moduli of the upper elastic actuator and lower bio-lamella were 3 and 1 MPa, respectively.

## 4. Results

### 4.1. Bending Deformation

The bio-toe was fixed on the mobile platform, and negative air pressure of −200 kPa and positive pressure varying from 20 to 70 kPa at 10 kPa intervals was applied in sequence. Figure 6a shows the bending states of the bio-toe under atmospheric, positive, and negative pressure, respectively. The bending curvature *C* of the four bio-toes versus pressure is shown in Figure 6b. According to the definition in Section 2.1, the curvature of the bio-toe is negative when bending toward the back and positive when bending toward the front. Under positive pressure, the curvature of all four toes increases approximately linearly with increasing pressure ($R_{unit1}^{2}$ = 0.9839, $R_{unit2}^{2}$ = 0.9828, $R_{unit3}^{2}$ = 0.9801, $R_{unit4}^{2}$ = 0.9833). It approaches zero at approximately 30 kPa, indicating that the bio-toes bend to a horizontal state. The difference in bending curvature between the four bio-toes is slight, with the bending deformation of bio-toe1 being slightly higher than that of the remaining three bio-toes.

### 4.2. Adhesion Performance

Figure 7a shows the procedure for testing the shear and normal adhesion performance of the four bio-toes. (1) Adhesion: the fixed bio-toe gradually bends toward the abdomen under positive pressure, contacting and adhering to the force-measuring platform. (2) Shear drag: a shear forward force is applied to the substrate when the bio-toe adheres to the substrate, which is similar to the simulation (Figure 4c), indicating that the bio-toe does not adhere stably. Therefore, the moving platform is controlled to drag the bio-toe horizontally toward its root at 0.1 mm/s and stop when the shear adhesion force reaches zero. (3) Shear/normal pull-off: the moving platform is controlled to move horizontally/vertically at 0.1 mm/s until the bio-toe separates from the force-measuring platform.

We measured shear/normal adhesion performance of the four bio-toes under positive air pressures varying from 20 to 70 kPa at 10 kPa intervals. Figure 7b,c show typical curves of tangential force, normal force, and contact area versus time for bio-toe3 in shear and normal adhesion performance tests, respectively. The behavior and mechanical properties during adhesion and shear drag of two tests were the same. The bio-toe adhered to the substrate, applying a shear forward force and a normal preload (defined as *F*_pre_) during adhesion. Meanwhile, the contact area increased rapidly from zero to a steady value (defined as *S*_ad_). The shear forward force returned to zero after shear drag. During shear pull-off (Figure 7b), the bio-toe made tight contact with the substrate, and the shear adhesion force increased rapidly and approximately linearly, whereas the normal contact force and contact area decreased very slowly. When the shear adhesion force reached its peak, the adhesion instantaneously failed (the shear adhesion force and the normal contact force at the moment before failure are defined as *F*_s_sp_ and *F*_n_sp_), and the contact area dropped to zero instantaneously. The shear force did not drop to zero instantaneously owing to the friction. According to the small graphs in Figure 7b, the contact areas of three bio-lamellae decreased simultaneously. During normal pull-off (Figure 7c), the shear adhesion force increased rapidly for a short period and then decreased slightly. This is because the two bio-lamellae near the root were released from the substrate, according to the small graphs in Figure 7c. Meanwhile, the contact area decreased to another stable contact state. After that, the shear adhesion force continued to increase as the normal contact force slowly decreased to a negative value, indicating that the interfacial contact state changed from compression to adhesion. When the adhesion force reached its peak, the adhesion failed instantaneously (the shear adhesion force and the normal contact force at the moment before failure are defined as *F*_s_np_ and *F*_n_np_).

Figure 8a,b show the results of the normal preload (*F*_pre_) and the area when it comes into adhesion (*S*_ad_), respectively. The *F*_pre_ of all four bio-toes increased steadily with air pressure in small increments and was stable within 1.2N overall. The difference in *F*_pre_ between the four bio-toes was slight but fluctuated greatly. The *S*_ad_ of all four bio-toes showed a trend of first increasing and then decreasing with the increasing air pressure. The *S*_ad_ of bio-toe3 was the highest throughout, peaking at 125 mm^2^ at 50 kPa. The greatest difference was at 50 kPa, where the *S*_ad_ of bio-toe3 was 9.4, 7.7, and 1.9 times greater than that of bio-toe1, 2, and 4, respectively.

Figure 8c,d show the results of the normal contact force (*F*_n_sp_) and shear adhesion force (*F*_s_sp_) at shear pull-off. The *F*_n_sp_ of all four bio-toes increased approximately linearly with the increasing air pressure from about 0.15 N at 20 kPa to about 1 N at 70 kPa ($R_{\mathrm{unit}1}^{2}=0.9903$, $R_{\mathrm{unit}2}^{2}=0.9675$, $R_{\mathrm{unit}3}^{2}=0.8449$, $R_{\mathrm{unit}4}^{2}=0.9524$). The difference in *F*_n_sp_ between the four bio-toes was slight but fluctuated widely. The *F*_s_sp_ of all four bio-toes showed a trend of first rising and then falling with the increasing air pressure. The *F*_s_sp_ of bio-toe3 was the highest throughout, peaking at 7.5 N at 50 kPa. The greatest difference was at 50 kPa, where the *F*_s_sp_ of bio-toe3 was 6.3, 2.7, and 1.7 times greater than that of bio-toe1, 2, and 4, respectively.

Figure 8e,f show the results of the normal contact force (*F*_n_np_) and shear adhesion force (*F*_s_np_) at normal pull-off. A force greater than 0 N means that the interface is compression, and the opposite is adhesion. The *F*_n_np_ of bio-toe3 and 4 was less than 0 N throughout, reaching the negative peak of −1.1 N at 40 kPa and −0.71 N at 60 kPa, respectively. The *F*_s_np_ of all four bio-toes showed a trend of first rising and then falling with the increasing air pressure. The *F*_s_np_ of bio-toe3 was the highest throughout, peaking at 4.3 N at 60 kPa and then falling back slightly. The *F*_s_np_ of bio-toe4 was second only to bio-toe3 throughout. The greatest difference was in the range of 40~60 kPa, where the *F*_s_np_ of bio-toe3 was 3.8, 2.8, and 1.5 times greater than that of bio-toe1, 2, and 4, respectively.

### 4.3. Active/Passive Release Performance

Figure 9a illustrates the test procedure for the active/passive release performance of the bio-toe. The procedures of the adhesion and shear drag are the same as those in Section 4.2. (1) Active release: the bio-toe bends toward the back rapidly under negative pressure of −200 kPa, releasing and separating from the force measuring platform. (2) Passive release: the bio-toe tends to return to its initial upward state under atmospheric pressure, away from the platform (the bio-toe cannot achieve 100% release success rate when passive releasing). We measured active and passive release performances of the four bio-toes with positive air pressure varying from 20 to 70 kPa at 10 kPa intervals in the adhesion procedure.

The typical curves of shear and normal forces versus time for bio-toe3 in the active and passive release performance tests are shown in Figure 9b and Figure 9c, respectively. The behavior and mechanical properties during adhesion and shear drag are similar to those in shear adhesion performance test (Figure 7b). During active release (Figure 9b), the bio-toe was subjected to −200 kPa pressure instantaneously and bent towards the back rapidly, generating a shear impulse force *F*_s_ar_ and a normal impulse force *F*_n_ar_, respectively. After that, the impulse forces fell back to zero quickly, indicating a successful release. In addition, there is more than one shear impulse force, indicating that the discrete bio-lamellae may not be released from the substrate synchronously. The normal impulse force decreased from positive to negative, indicating an interfacial normal adhesion generated at the moment of release.

During passive release, the bio-toe tended to return to its initial upward state, with rebound forces pulling the bio-lamellae away from the substrate. Only the bio-lamellae near the end of the bio-toe can release from the substrate, generating a shear impulse force *F*_s_pr_ and a normal impulse force *F*_n_pr_, respectively. Due to the lack of active upward bending deformation, the adhesion force of the root bio-lamella was larger than the rebound force, resulting in failure to release from the substrate.

We define the active release force *F*_ar_ of the bio-toe as the combined force of *F*_s_ar_ and *F*_n_ar_, and the passive release force *F*_pr_ as the combined force of *F*_s_pr_ and *F*_n_pr_. The *F*_ar_ and *F*_pr_ of all four bio-toes (Figure 10) showed a trend of first increasing and then gently decreasing with the increasing air pressure. The greatest difference in *F*_ar_ and *F*_pr_ was in the range of 50–70 kPa, where both *F*_ar_ and *F*_pr_ of bio-toe2 reached about 2 N, which was 1.5–2 times higher than that of bio-toe3 and 4. The *F*_ar_ of bio-toe1 and 2, as well as the *F*_pr_ of bio-toe2, were significantly affected by air pressure. In contrast, the *F*_ar_ of bio-toe3 and 4 was stable in the range of 0.5 N to 1.2 N overall.

We also used statistics on the release success rate of the active and passive release, respectively. The release success rates of active release of the four bio-toes were all 100%. However, the release success rate of passive release of bio-toe1, 2, 3, and 4 were 100%, 87%, 0%, and 0%, respectively. For bio-toe2, the air pressure range for successful release was 50–70 kPa.

### 4.4. Application in Robotic Gripper and Climbing Robot

The bio-toes can be used as basic adhesive units in a robotic gripper or at the end of a wall-climbing robot. In this study, we placed two bio-toes oppositely on a parallel frame to assemble an adhesive gripper, which can control the opening angle of the pair of two bio-toes according to the size of the target. In the gripping test (Figure 11a), dozens of objects used in daily life and large smooth hemispheres (difficult to gripped by conventional friction grippers [26,30] were randomly selected as gripped targets, covering rigid and flexible objects, even soft fluid objects. This adhesive gripper demonstrated versatility. Relying on the contact adaptability of bio-toes, the gripper could effectively fit targets with cylindrical surfaces, spherical surfaces (smooth hemispheres with radius from 75 mm to 150 mm), right-angle surfaces, irregular surfaces (cups, mice), and soft surfaces (soft snack bag and water bag) and achieve stable adhesive gripping without damaging or transitionally squeezing the surface. Not only that, but the gripper was capable of gripping weights of 2 kg, showing a good load capacity.

We integrated the adhesive gripper into an automatic adhesive gripping system, which can identify and locate the target through a vision module and drive the adhesive gripper to the appropriate position to grip the target and place it at the specified location through a robot arm (Figure 11b). This adhesive gripping system, with the good inherent flexibility of the soft bio-toes and the large stable adhesive space of the adhesive gripper, could realize the reversible adhesive gripping and release of the large curved surface, flexible and easily deformed snacks, fragile eggs, etc. The automatic adhesive gripping system demonstrated good interaction safety and gripping diversity and stability. In the future, it can be applied to agricultural picking, industrial grasping, medical rehabilitation, and other fields.

We designed an adhesive foot consisting of four bio-toes with tabular bio-lamellae that could be used as the end foot of the wall-climbing robot (Figure 11c). The adhesive foot could generate a peak normal adhesion force of 3.8 N in the normal pull-off test. The quadrupedal wall-climbing robot equipped with four adhesive feet was capable of climbing on a large smooth curved surface in a weightless condition (with a weight equal to the robot’s own weight suspended from a fixed pulley at the top to counteract the robot’s weight). With the flexible and controllable bending ability and good adaptability of the bio-toes, the adhesive feet could actively envelop and adhere to non-flat surfaces to assist the robot move stably. The autonomous adhesion/release of the adhesive feet could greatly simplify the robot’s leg moving trajectory; therefore, the adhesive feet could be flexibly integrated into the robot’s multiple gait patterns to enable the robot to quickly switch between stable adhesion and efficient release when crawling on the curved surface. The technology could be applied to the specialized robot and the capture of non-cooperative targets in space.

## 5. Discussion

### 5.1. Reversible Adhesion Performance

The bending curvatures of the bio-toes are linearly and positively correlated with positive air pressure, indicating that the bending deformation is continuous and controllable. Compared to rigid [6,20] and underactuated [5,24,25,26] adhesive units, the bio-toes can actively envelop a wider range of targets with flexibility. The upward bending of the bio-toe driven by negative pressure is necessary for successful release, especially for bio-toes with excellent adhesion ability (bio-toe3 and 4). Together with the rebounding force from the initial upward structure, the upward bending behavior can generate a sufficient detaching force to overcome the adhesion force for complete release (100% success rate of release). The idea of increasing the upward bending angle of the elastic actuator by optimizing the shape of the cavity top [33] is also applicable to bio-toes. The shear adhesion performances of bio-toes are better than the normal, which is similar to that of the gecko-inspired tilted adhesive pad attached to a curved beam [14,15], so the shear sliding motion can help to enhance the adhesion state.

The adhesion performance of the bio-toes proved that the gecko-inspired bio-hierarchical structure significantly improves the bio-toe performance. The specific effects are as follows: (1) The titled tabular and curved bio-lamellae act as soft springs to sustain most of the normal deformation, in a similar manner to gecko lamellae. They have a larger distance range of low squeeze contact. In particular, the tabular bio-lamella achieved a 60% contact rate at a low preload of 0.5 N, which effectively compensates for the low contact adaptability of bio-toe1. An adhesive pad with titled structure was also proposed and proved to be robust in terms of excess vertical compression [15]. A similar bionic lamella structure [31] was proposed to enhance contact adaptation to nonplanar surfaces, but how this bionic lamella might be integrated with the supporting/driving layer to form a controllable and reversible adhesive unit was not investigated further. (2) The tabular and curved bio-lamellae provide the bio-toe with greater robustness to cope with changes in its own posture and disturbances from the external environment owing to the inherent flexibility of the soft material and the lower contact stiffness of the bionic lamellar-shaped structure. The results indicated that the tabular bio-lamella could maintain a stable 60% contact rate despite a ±10° tilt in contact posture, whereas that of the square bio-lamella was reduced to almost zero, which not only validates the rationality of imitating the biological hierarchical structure described in Section 2.2, but also demonstrates the soft spring effect of the gecko lamellae from an engineering perspective [17]. (3) Discrete bio-lamellae can hinder the expansion of interfacial adhesion failure cracks, avoid instantaneous desorption, and enhance adhesion stability. In the normal adhesion performance test in Section 4.2, Figure 7c shows that the two bio-lamellae near the root were the first to release from the substrate, and the bio-lamella near the end was unaffected and continued to remain adherent. The final shear adhesion force at complete release was approximately three times higher than that at the first release of the two bio-lamellae. In addition, the pull-off distance at complete release was approximately 6 cm, which was six times greater than that at the first release of the two bio-lamellae, thus significantly widening the stable adhesion space.

Compared to existing -toe3 in the shear adhesion performance test were all maintained above 5, and they even reached approximately 12 (approximately 5 and 3 times higher than those of bio-toe1 and 2, respectively) under a pressure of 40–50 kPa. In contrast, the ratio of the shear friction force to the preload of the conventional frictional gripping unit was less than 1 [32], indicating that the adhesive bio-toe is capable of lifting heavy objects without applying excessive squeezing force and is very suitable for grasping soft, fragile targets (such as eggs and water bags). In the adhesion performance test, the ratios of normal (shear) adhesion force to the preload of bio-toe3 and 4 could reach approximately 1.7 (5.5) and 1 (4), respectively. The soft pneumatic adhesive pad proposed by Tian et al. [37] can adhere to rough surfaces, but it requires more than 30 mN of a normal preload to generate a normal adhesive force of 10.2 mN, implying a ratio of normal adhesion force to preload of less than 0.33. The soft adhesion-based gripping system proposed by Song et al. [38] has a ratio of normal adhesion force to a preload of approximately 5, but it lacks maneuverability owing to the lack of a controllable supporting layer and sufficient structural stiffness to apply moments to the target.

### 5.2. Applications

Adhesive gripping is an emerging robotic gripping method that has a wider range of applications than traditional friction and vacuum adsorption gripping [39]. The adhesive gripper in this study showed both good load capacity (2 kg weight) and excellent gripping versatility, which are essential for the gripper’s application in multiple fields. The gripping of large smooth surfaces has been a challenge for conventional friction grippers [26], and interfacial adhesion technology is a feasible solution [26,30,32]. Most existing adhesive grippers are rigid and only applicable to flat surfaces [6,19,20,21]. Grippers that integrate electrostatic adhesion and gecko-inspired dry adhesion can grip large curved surfaces, but they require high-voltage actuation [24]. To grip large smooth targets, the adhesive gripper proposed in this study was able to envelop smooth spherical surfaces with a curvature radius of 75–150 mm owing to its flexible and controllable bending capability, and it relied on the soft adaptable adhesive layer (bio-lamellae) to fully adhere to the non-flat surface. To grasp soft and fragile objects, it is important to avoid squeezing and damaging the target surface. A gripper made of soft materials [32] or an embedded contact-sensing module [40] can alleviate this problem, but still cannot overcome the fact that the higher shear friction of frictional grasping depends on higher normal squeezing. Because the dry-adhesive surface can generate stable interfacial adhesion with a low preload, the adhesive gripper can grasp soft objects (fruits, snacks, vegetables, etc.) while keeping them undeformed. This adhesive gripping technology could be applied in the future in agricultural harvesting [41]. Soft fluidic objects cannot be grasped stably even when sufficient squeezing pressure is applied, owing to their uncontrollable shape. Enveloping the fluidic object to form a shape closure can realize the grasping, but the size of the gripper increases with the volume of the fluidic body. We propose the idea of relying on dry adhesion or similar interfacial adhesion techniques [42] to attach part of the epidermal region of a fluidic body to extract the entire fluidic body and successfully grasp a water bag with a mass of 500 g without enveloping.

Wall-climbing robots are used in special conditions and aerospace applications, such as post-disaster rescue and aerospace repair, owing to their excellent locomotion abilities. Most existing wall-climbing robots use rigid flat adhesion units for their feet, which cannot actively adhere and release. They rely on cam separation mechanisms [22], motor-driving legs to generate trajectory-based oscillations [23], or rotations [9] for adhesion and release, which increases the complexity of the leg structure and burdens the planning of the adhesion and release trajectories. In addition, this robot can climb on vertical surfaces or even negatively inclined surfaces but is not suitable for curved surfaces. Robotic adhesive feet based on bio-toes have good adaptability and autonomous reversible adhesion/release ability, which helps the wall-climbing robot to climb freely on a smooth curved surface in a weightless environment. The adhesive foot can effectively and controllably switch between adhesion and release, thus eliminating the need for additional trajectory planning by the robot legs for adhesion and release, reducing the control cost and improving the working efficiency of the robot system. With inherent flexibility, the adhesive foot can effectively reduce internal impact and external interference during the climbing process, thereby improving the stability of the robot system.

### 5.3. Mathematical Model and FEA Simulation

The mathematical model can qualitatively analyze the effects of geometric parameters such as cavity width, cavity height, cavity wall thickness, and thickness of the inextensible layer (abdomen) on the adhesion performance of the elastic actuator. The accuracy of the mathematical model is guaranteed by fitting the strain–stress relationship of the material. This function describes the nonlinear properties of the material and can be used for curve control; however, it does not provide an explicit interpretation of the material model, which is a challenge in large-deformation theory [33].

The Mooney–Rivlin hyperelastic material model used in FEMs has previously been employed to describe the mechanical properties of rubber [30,34]. The simulation results of the bending deformation of the bio-toes were within a 10% error of the experimental results, which demonstrates the rationality of the FEMs. However, the main reasons for this 10% error may be (1) mechanical errors in mold manufacturing, (2) assembly errors and testing errors of pressure sensors during experimental testing, and (3) simulation errors, such as component type, mesh density, and material properties. The simulation results and mathematical model are consistent, indicating that higher, wider, and thinner cavities contribute to the bending deformation of the elastic actuator. In addition, the FEMs also predict the effect of the structural geometry parameters on the contact force of the elastic actuator and find that the contact force and deformation are highly consistent, which guides the determination of the structural parameters of the bio-toe. In the simulation of adhesion contact, the cohesive zone model (CZM) was used to describe the interfacial adhesion and failure behavior of the bio-toe during adhesion, because it can simulate the initial uncontacted but in-process adhesive contact conditions [43,44]. However, the model setup is demanding and difficult to converge. Some studies simulated the interface adhesion behavior by establishing interfacial spring constraints [45,46], which have more intuitive and controllable parameter settings, but can only simulate the case of release from the substrate with an initial adhesion state. The FEMs of the four bio-lamellae in this study are mainly used to analyze the behavior and mechanical properties when contacting the substrate and do not consider the release process after contact; thus, the interfacial contact properties are defined as smooth and frictionless. The simulation results demonstrate the superior contact adaptation and robustness of the tabular bio-lamella, which is consistent with the experimental results presented in Section 4.2.

## 6. Conclusions

This work proposes a hierarchical bio-toe inspired by the sophisticated structural propertity and reversible adhesion motility of gecko toes. Theoretical and experimental results demonstrated that the multilayered structure provides bio-toes with gecko-like performance. The bio-toe can flexibly and controllably adapt to the shape of a wide range of targets by relying on the upper supporting/driving layer, and it can adhere to the target stably with adaptability and maintain high robustness against interference by relying on the lower adhesion layer. The synergy of the upper and lower layers enables the bio-toe to adhere to or release from targets autonomously. In applications, an adhesive gripper consisting of a pair of bio-toes demonstrates good gripping versatility and loading capacity. The wall-climbing robot with bio-toes as end feet shows stable and reversible switching of adhesion and release, as well as good compatibility with complex automation systems. Furthermore, the adhesive bio-toe can also be used for human–robot interaction, agricultural harvesting, and aerospace noncooperative target capture. In addition, this study has implications for the bionic implementation of adhesion technology in engineering, in terms of methodological strategies.

## Figures and Tables

**Figure 1 biomimetics-08-00040-f001:**
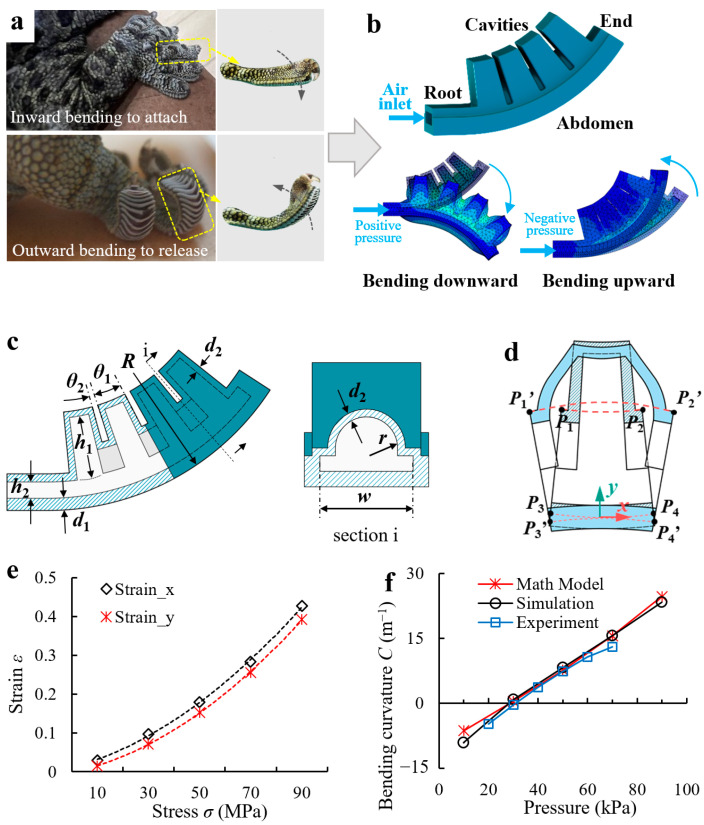
(**a**) Reversible adhesive behavior of the gecko toe. (**b**) Gecko-inspired fluidic-driven elastic actuator acting as the supporting/driving layer of the bio-toe and its upward/downward bending deformation when driven by positive/negative pressure. (**c**) Schematic of the elastic actuator. (**d**) Illustration of the mathematical model of the bending deformation. (**e**) Fitting functions of the strain–stress relationship for the elastic actuator on the *x* and *y* axis. (**f**) Bending curvatures of the elastic actuator versus pressure from the mathematical model, simulation, and experimental results.

**Figure 2 biomimetics-08-00040-f002:**
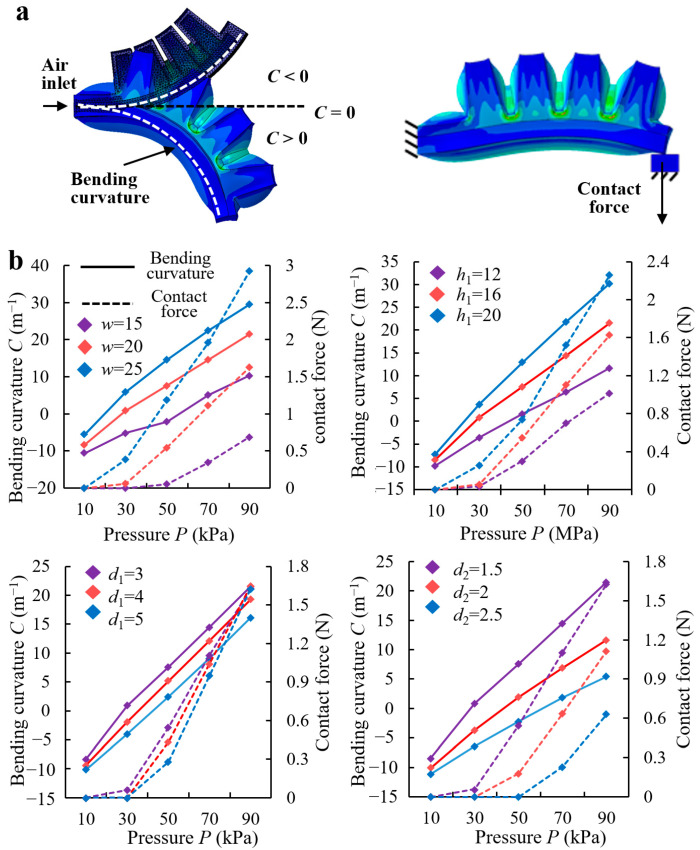
(**a**) Finite element models of the elastic actuator’s bending deformation and contact force under pressure drive. (**b**) Simulation results of the effects of width of the chamber *w*, height of the chamber *h*_1_, thickness of the inextensible layer *d*_1_, and thickness of the chamber *d*_2_ on the bi-directional bending curvature and contact force of the elastic actuator under pressure drive.

**Figure 3 biomimetics-08-00040-f003:**
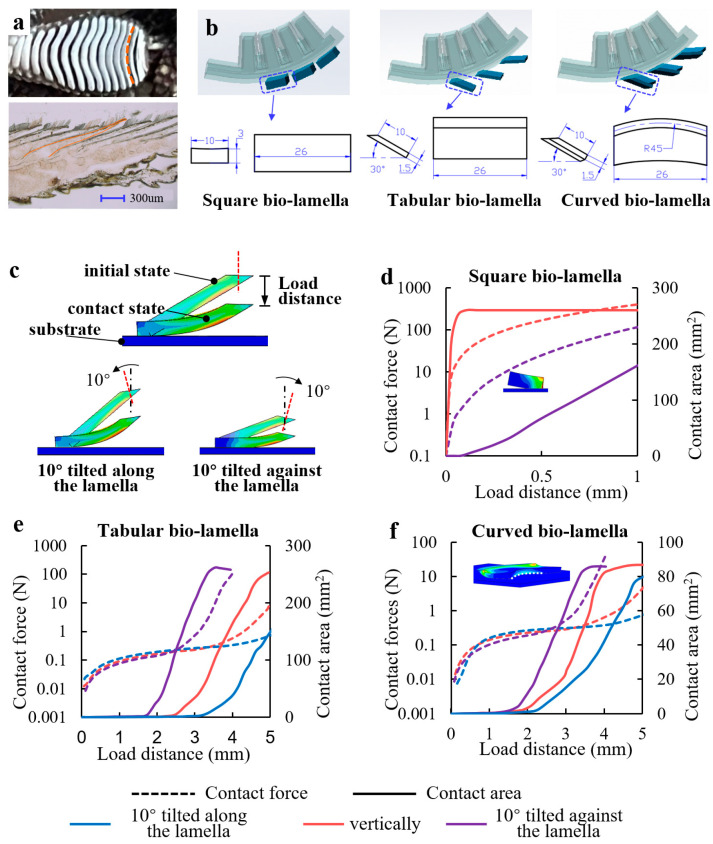
(**a**) Longitudinal frozen slicing and bottom view of the gecko toe’s lamellae. (**b**) Three gecko-inspired bio-lamellae: square, tabular, and curved bio-lamella. (**c**) Finite element models of the tabular bio-lamella’s behavior and mechanical characteristics when contacting a flat substrate under three contact modes (vertically, 10° tilted along the lamella, and 10° tilted against the lamella). Simulation results of contact force and contact area versus load distance for the (**d**) square, (**e**) tabular, and (**f**) curved bio-lamellae.

**Figure 4 biomimetics-08-00040-f004:**
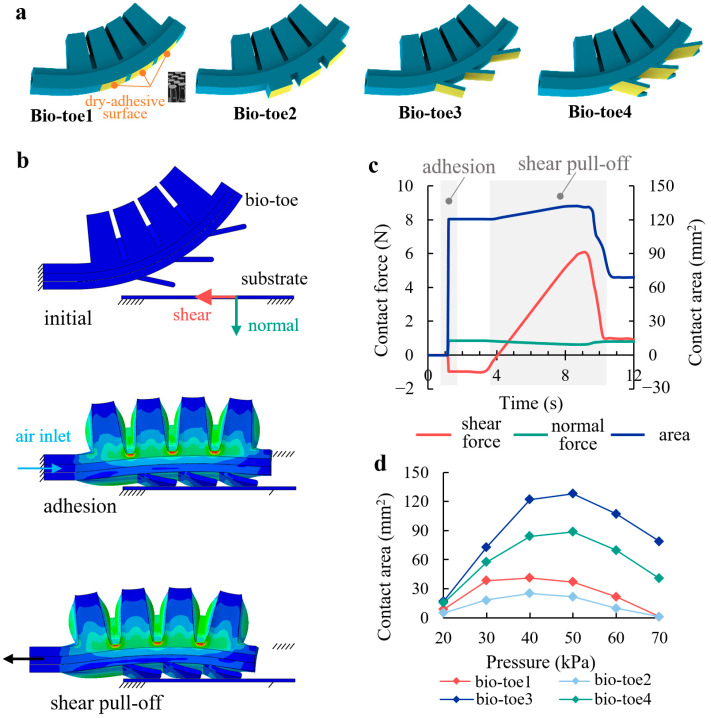
(**a**) Four types of bio-toes. (**b**) Finite element model of bio-toe3 during adhesion to and shear pull off from the flat surface. (**c**) Simulation results of the contact behavior and mechanical properties. (**d**) Contact areas of the four bio-toes in adhesion versus applied pressure.

**Figure 5 biomimetics-08-00040-f005:**
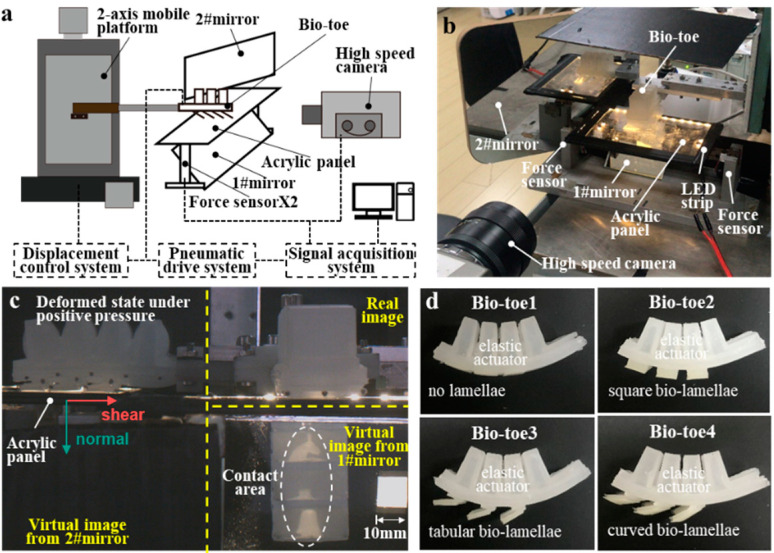
(**a**) Diagram of the synchronous testing platform of adhesive contact state and mechanics. (**b**) Synchronous testing platform. (**c**) A frame from the video captured by the high-speed camera. (**d**) Prototypes of the four bio-toes.

**Figure 6 biomimetics-08-00040-f006:**
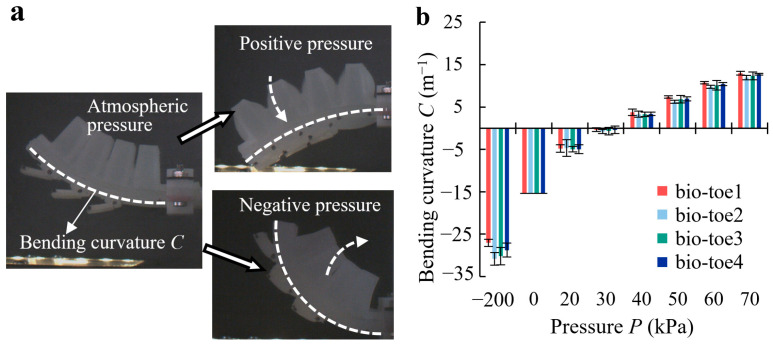
(**a**) State of the bio-toe under atmospheric, positive, and negative pressure. (**b**) Bending curvature of the four bio-toes versus pressure.

**Figure 7 biomimetics-08-00040-f007:**
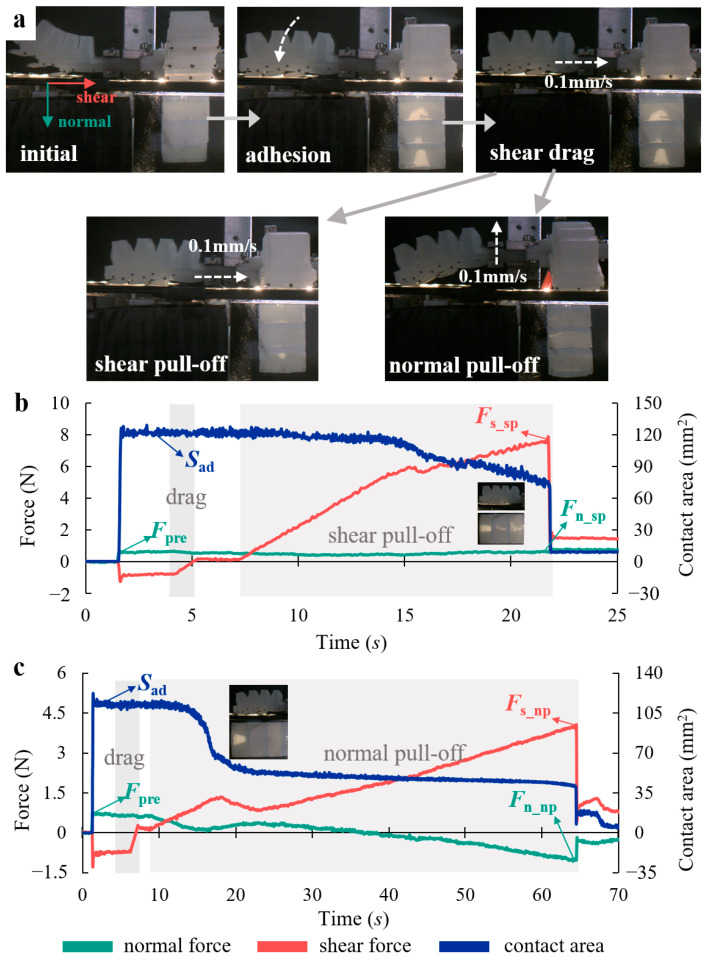
(**a**) Flow of the adhesion performance test of the bio-toe. Typical curves of shear force, normal force, and contact area versus time for the bio-toe in the (**b**) shear and (**c**) normal adhesion performance tests.

**Figure 8 biomimetics-08-00040-f008:**
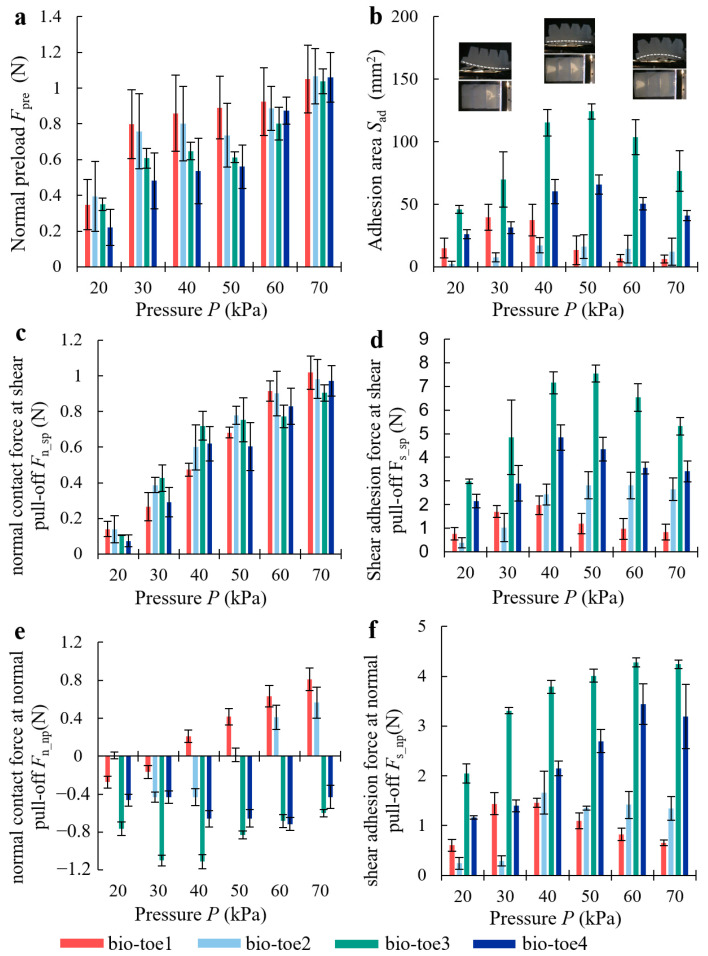
(**a**) Normal preload *F*_pre_ and (**b**) adhesion contact area *S*_ad_ of the four bio-toes during the adhesion performance tests. (**c**) Normal adhesion force *F*_n_sp_ and (**d**) shear adhesion force *F*_s_sp_ of the four bio-toes during the adhesion performance test of shear pull-off. (**e**) Normal adhesion force *F*_n_np_ and (**f**) shear adhesion force *F*_s_np_ of the four bio-toes during the adhesion performance test of normal pull-off.

**Figure 9 biomimetics-08-00040-f009:**
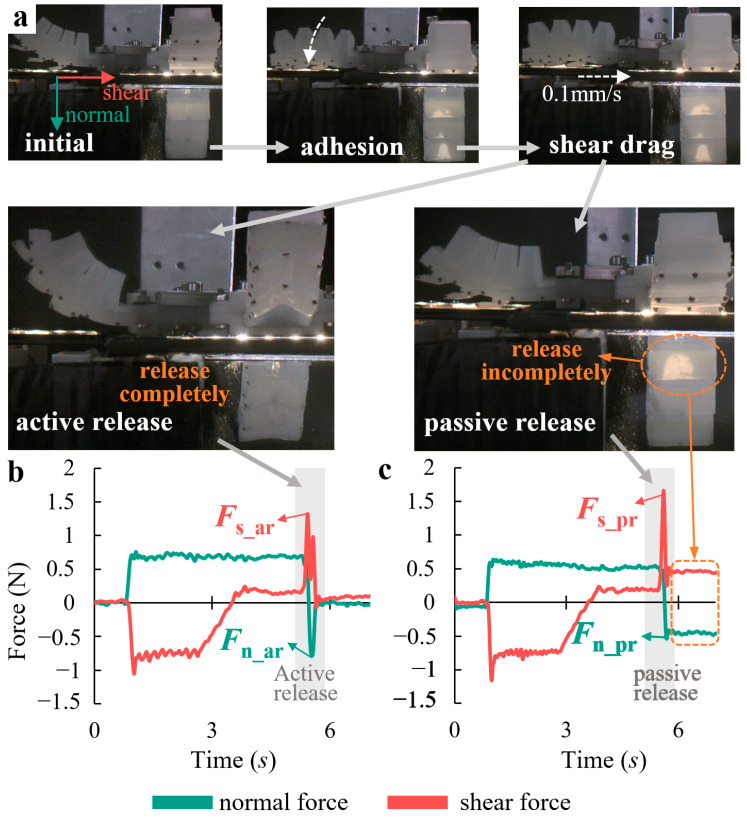
(**a**) Flow of the active/passive release performance test of the bio-toe. Shear and normal forces of the bio-toe during the (**b**) active and (**c**) passive release test.

**Figure 10 biomimetics-08-00040-f010:**
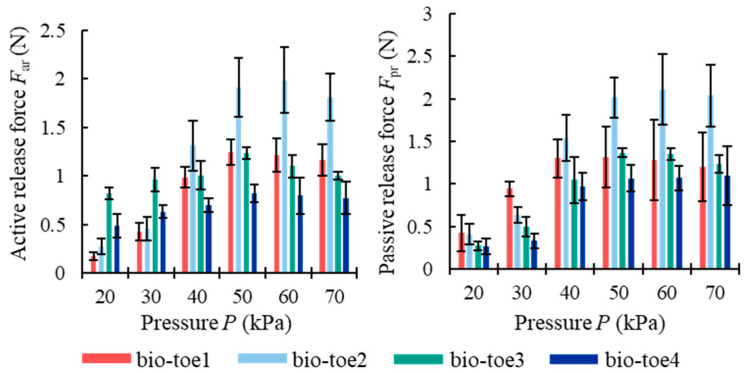
Active and passive release force of the four bio-toes.

**Figure 11 biomimetics-08-00040-f011:**
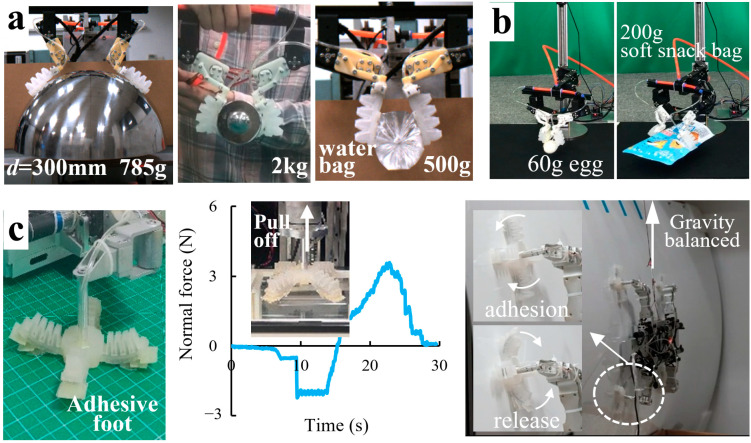
Application of the bio-toe in (**a**) a robotic adhesive gripper, (**b**) an automatic adhesive gripping system, and (**c**) a wall-climbing robot.

**Table 1 biomimetics-08-00040-t001:** Values of structural parameters of the elastic actuator.

Structural Parameters	Values (mm)
$\mathrm{Height}\;\mathrm{of}\;\mathrm{the}\;\mathrm{chamber}\;h_{1}$	16
$\mathrm{Height}\;\mathrm{of}\;\mathrm{the}\;\mathrm{airway}\;h_{2}$	4
Width of the chamber w	20
$\mathrm{Thickness}\;\mathrm{of}\;\mathrm{the}\;\mathrm{inextensible}\;\mathrm{layer}\;d_{1}$	3
$\mathrm{Thickness}\;\mathrm{of}\;\mathrm{the}\;\mathrm{chamber}\;d_{2}$	1.5
$\mathrm{Center}\;\mathrm{angle}\;\mathrm{of}\;\mathrm{the}\;\mathrm{chamber}\;\theta_{1}$	10
$\mathrm{Center}\;\mathrm{angle}\;\mathrm{of}\;\mathrm{the}\;\mathrm{rib}\;\theta_{2}$	2
Radius of the actuator R	65
Radius of the rib r	6

## Data Availability

The data generated and/or analyzed during the current study are not publicly available for legal/ethical reasons but are available from the corresponding author upon reasonable request.

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
