# Peer review of "Reversible Adhesive Bio-Toe with Hierarchical Structure Inspired by Gecko"

_biomimetics, 2023, doi:10.3390/biomimetics8010040_

Round 1

Reviewer 1 Report

The authors developed gecko-inspired hierarchical bio-toe and tested the adhesion/release  performances. They applied the developed system to grippers and wall-climbing robots.

The motivation is sound, the manuscript is well written, and their results are interesting and reasonable. Once the following point is considered and modified, the reviewer will recommend the manuscript for acceptance in Biomimetics.

The following is a comment from the reviewer.

-       In Figure 1, the image of the longitudinal section is not so clearly shown, and it is difficult to see how the inward/outward bending contribute to attachment/detachment from the two high-speed pictures. The authors should replace the images to clearly show structure and motions of gecko feet.

-       Recently, the analyses on the tilted pad during normal compression/debonding with lateral pull/push were conducted experimentally and theoretically in the papers below. The authors should cite the paper to introduce the past results and to strengthen their discussion.

B. Zhao, N. Pesika, H. Zeng, Z. Wei, Y. Chen, K. Autumn, K. Turner, J. Israelachvili, Role of tilted adhesion fibrils (setae) in the adhesion and locomotion of gecko-like systems. The Journal of Physical Chemistry B, 113(12), pp.3615-3621 (2009).

T. Yamaguchi, A. Akamine, Y. Sawae, On/off switching of adhesion in gecko-inspired adhesives, Biosurface and Biotribology 7 (2), pp.83-89 (2021).

Author Response

Thank you very much for your professional review. We have carefully read the comments and suggestions, and the point-to-point responses are appended.

Reviewer 2 Report

The article is devoted to practical construction of the devices, which mimic an ability of adhesive animals to fast locomotion. It is well known, that such an ability is related to the both feature: hierarchical feet structure and to the reversible adhesion motility. According to this knowledge a design and study of the structure–function relationship of the artificially build adhesive devises is very interesting from both points of view: it can help to understand biologic adhesive systems and can help to create practical bionic applications.

In the manuscript a study, of real prototype of such a device is presented. It combines a reversible adhesion/release behavior and structural properties of gecko toes investigation with the study of a hierarchical adhesive bionic toe. The system consists of upper elastic actuator as the supporting/driving layer and lower bionic lamellae as the adhesive layer, which can adhere to and release from the surfaces reversibly when driven by changes (positive and negative) of pressure. The conceptual view of the systems, as well as initially motivating pictures of the working gecko toe, calculated and numerically found dependencies, are shown in Figs. 1-2 of manuscript.

Preliminary the authors apply a mathematical model of the nonlinear deformation and a finite element model of the adhesive contact of the bio-toe. After they perform a set of the experimental tests, concentrated on elucidation of the effects of the structure and actuation on the adhesive behavior and mechanical properties of the bio-toe. Further, the authors reproduce experimental setup, as well as experimentally found results, which are summarized in Figs. 3- 10, including some possible applications in Fig.11.

In particular, one can mention that the authors have found that the upward bending of the bio-toe under negative pressure provided sufficient rebounding force for 100% success rate of release and high ratio of shear adhesion force to preload (approximately 12), which is higher than that of most existing adhesion and frictional gripping systems. In general, the proposed in the manuscript bio-toe shows good adaptability, load capacity, and reversibility of adhesion.

To my mind, the proposed by authors reversible adhesive bio-toe can be successfully applied in different fields, including space, industry, etc. The article is well written, its idea is absolutely clear and the results are well supported by wide set of the numerical and experimental results. I recommend it to the publication in present form.

Author Response

Thank you very much for your recognition of our research, and we will continue to explore in the field of bionic adhesion technology and its applications.